# Brianolide from *Briareum stechei* Attenuates Atopic Dermatitis-like Skin Lesions by Regulating the NFκB and MAPK Pathways

**DOI:** 10.3390/biom15060871

**Published:** 2025-06-14

**Authors:** Chia-Chen Wang, Kang-Ling Wang, Yu-Jou Hsu, Chao-Hsien Sung, Mei-Jung Chen, Meng-Fang Huang, Ping-Jyun Sung, Chi-Feng Hung

**Affiliations:** 1School of Medicine, Fu Jen Catholic University, New Taipei City 24205, Taiwan; jamiewang@tma.tw (C.-C.W.); 413138010@mail.fju.edu.tw (M.-F.H.); 2Department of Dermatology, Cardinal Tien Hospital, New Taipei City 23148, Taiwan; 3Division of Metabolism and Endocrinology, Department of Internal Medicine, Taoyuan Armed Forces General Hospital, Taoyuan 32551, Taiwan; klwang@aftygh.gov.tw; 4PhD Program in Pharmaceutical Biotechnology, Fu Jen Catholic University, New Taipei City 24205, Taiwan; 411138028@m365.fju.edu.tw (Y.-J.H.); a02664@mail.fjuh.fju.edu.tw (C.-H.S.); 5Division of Anesthesiology, Fu Jen Catholic University Hospital, Fu Jen Catholic University, New Taipei City 24352, Taiwan; 6Institute of Biomedical Engineering, School of Health and Medical Engineering, Ming Chuan University, Taoyuan 33300, Taiwan; mjchen@mail.mcu.edu.tw; 7National Museum of Marine Biology and Aquarium, Pingtung County 94450, Taiwan; pjsung@nmmba.gov.tw; 8Department of Marine Biotechnology and Resources, National Sun Yat-Sen University, Kaohsiung 80424, Taiwan; 9School of Pharmacy, Kaohsiung Medical University, Kaohsiung 80708, Taiwan

**Keywords:** *Briareum stechei*, brianolide, soft coral, atopic dermatitis, inflammatory

## Abstract

Atopic dermatitis (AD) is a common chronic skin disease affecting both children and adults. Currently lacking a clinical cure, AD presents significant physical and emotional challenges for patients and their families, substantially impacting their quality of life. This underscores significant unmet needs in AD management and highlights the necessity for developing effective therapeutic applications. Recently, several chlorine-containing active substances with promising pharmacological activity have been discovered in soft corals cultivated through coral farming. Among these, brianolide, isolated from the soft coral *Briareum stechei*, has shown promising potential. This study investigated brianolide’s regulatory effects on the inflammatory response in atopic dermatitis and its underlying mechanisms. Using an in vitro human keratinocyte cell line (HaCaT) stimulated with tumor necrosis factor-α (TNF-α)/interferon-γ (IFN-γ) to mimic AD inflammation, brianolide was found to inhibit cytokine and chemokine expression via the mitogen-activated protein kinase (MAPK) and nuclear factor kappa-light-chain-enhancer of activated B cell (NFκB)-signaling pathways. In an in vivo animal model of 2,4-Dinitrochlorobenzene (DNCB)-induced AD, brianolide demonstrated anti-inflammatory effects, reducing transepidermal water loss (TEWL), ear thickness, erythema, and epidermal blood flow. These findings provide new insights into brianolide’s activity against AD-related inflammation, elucidate potential mechanisms, and contribute to understanding the pharmacological potential of natural coral products for AD treatment.

## 1. Introduction

Atopic dermatitis (AD) is a chronic inflammatory skin disease that significantly impacts patients’ quality of life, affecting their daily activities, work efficiency, and overall health [1,2,3]. It is estimated that global prevalence of AD is 2.6% and affect 204.05 million people worldwide [4]. AD is characterized by eczema, dry skin, itching and erythema [5,6,7,8]. The key pathological mechanism of AD is disruption of skin barrier. Microorganisms, allergens and irritants could easily enter disrupted skin barriers [8]. Upon stimulus, epidermal cells release pro-inflammatory cytokines such as tumor necrosis factorα (TNF-α), Interleukin-1β (IL-1β), IL-6, IL-8, leading to persistent skin inflammation and amplification of immune response [9,10]. Long lasting symptoms may lead to secondary symptoms such as ulceration, pigmentation and skin lichenification [11,12]. The exact cause of AD remained unknown. Genetic factors, autoimmune dysfunction, environmental hazard, and psychological stress were proposed as possible precipitating factors of AD [13]. The chronic and recurrent nature of the disease imposes a considerable burden on individuals, families, and society [14]. Many patients express dissatisfaction with existing treatments, primarily due to side effects and inadequate efficacy. Common treatments, such as topical steroids and oral antihistamines, often fail to meet expectations for long-term relief and effective symptom management [15,16]. Additionally, poor treatment compliance in patients with AD often leads to poor treatment response. Newly developed medications, including biologics, are also prohibitively expensive. As a result, patients frequently switch or discontinue treatments, highlighting a significant unmet need in the management of atopic dermatitis [17].

Humans have long utilized natural substances derived from plants, animals, minerals, and other environmental resources to treat diseases [17,18,19]. Many traditional herbal remedies, passed down through generations, carry both cultural and medicinal significance. Consequently, natural products have gradually gained recognition in clinical research, with scientific evidence supporting their therapeutic benefits [20]. Many marine-derived compounds (such as polysaccharides, alkaloids, or marine biological extracts) exhibit anti-inflammatory, immunomodulatory, and barrier repair effects, suggesting potential benefits for AD patients. Marine-derived ingredients have been studied as potential adjuvant treatments for AD, with a variety used in topical ointments or oral supplements [21,22]. Research indicates that marine-derived extracts and isolated compounds, such as excavatolide B (EXCB), play crucial roles relevant to the pathological mechanisms of atopic dermatitis and present significant therapeutic potential [23,24,25]. Epi-oxyzoanthamine was isolated from zoanthid and could inhibit TNF-α/interferon-γ (INF-γ) expression in human keratinocytes (HaCaT) cells and reduced the phosphorylation of MAPK and NF-κB pathways [26]. Brianolide, a compound derived from the soft coral *Briareum stechei* [27,28], was investigated. Brianolide shares the same briarane skeleton as excavatolide, but its structure is distinctly different (Figure 1). This study employs two experimental approaches: an in vitro human keratinocyte cell line (HaCaT) model stimulated with TNF-α and IFN-γ to simulate eczema-like inflammation, for assessing the anti-inflammatory effects of brianolide; and an in vivo mouse model induced with DNCB to provoke AD-like inflammation.

## 2. Materials and Methods

### 2.1. Brianolide and Cell Line

The compound brianolide, a white powder with a purity exceeding 98%, was extracted from the soft coral *Briareum stechei* by Professor Ping-Jyun Sung of the National Museum of Marine Biology and Aquarium in Taiwan, as described in previous publications [28]. The human keratinocyte cell line HaCaT was provided by Dr. Nan-Lin Wu of the Dermatology Department at Mackay Memorial Hospital in Taiwan.

### 2.2. Cell Culture

The study was conducted using the HaCaT cell line. The cells were cultured in DMEM medium containing 10% fetal bovine serum and 1% antibiotics. The cells were incubated in a 37 °C cell culture incubator with 5% CO_2_ until they reached 90% confluence, at which point they were passaged for further culture.

### 2.3. MTT Assay

Cells were seeded into a 24-well plate at a density of 5 × 10^4^ cells per well. After a 24 h treatment with brianolide, 300 μL of MTT solution [3-(4,5-dimethylthiazol-2-yl)-2,5-diphenyl-2H-tetrazolium bromide] was added to each well. The plate was incubated at 37 °C for approximately 2 h. Following incubation, the medium was removed, and the resulting formazan crystals were dissolved in DMSO. The mixture was thoroughly homogenized, and cell viability was determined by measuring absorbance at 550 nm using an ELISA reader.

### 2.4. Quantitative Polymerase Chain Reaction (qPCR)

For mRNA expression analysis, HaCaT cells were seeded in 3.5 cm culture dishes until they reached 90% confluence. After growing for 24 h, cells were pre-treated with brianolide for 1 h, followed by stimulation with TNF-α/IFN-γ for 1 h. The cells were then harvested by scraping, centrifuged (16,000 rpm, 10 min, 4 °C), and the supernatant was collected.

RNA was extracted and purified using a total RNA isolation kit (GeneDireX^®^, Vegas, NV, USA). cDNA synthesis was performed according to the protocol of the purchased iScript™ cDNA Synthesis Kit (BIO-RAD, Hercules, CA, USA) to convert RNA into cDNA.

For quantitative PCR, PowerUp™ SYBR™ Green Master Mix (Applied Biosystems™, Waltham, MA, USA) was used. The reaction mixture included 7.5 μL of dd water, 0.25 μL of each forward and reverse primer, and 10 μL of SYBR GREEN. Quantitative PCR was conducted using the ABI StepOnePlus™ Real-time PCR System. Human primer sequences for RT-qPCR are listed in Table 1.

### 2.5. Western Blot Assay Analysis

The Western blot assay analyzes the changes in various proteins within cells through electrophoresis. HaCaT cells were seeded in 3.5 cm culture dishes and allowed to grow until 90% confluent. After resting for 24 h, the cells were pre-treated with brianolide for 1 h, followed by stimulation with TNF-α/IFN-γ for 30 min (for p38, ERK and JNK) or 1 h (for IκB). The cells were then harvested by scraping, transferred to a centrifuge tube, and sonicated for lysis. Samples were centrifuged at 13,200 rpm for 10 min at 4 °C, and the supernatant was collected for protein quantification using the Pierce protein assay kit (Pierce, Rockford, IL, USA).

A total of 20–40 μg of protein was subjected to electrophoresis using a 10% SDS-polyacrylamide gel, followed by electroblotting onto a PVDF membrane. After blotting, the PVDF membrane was placed in a solution of TBS-T (Tris-buffered saline/0.05% Tween 20) containing 5% non-fat dry milk and shaken for 1 h. The membrane was then washed three times with TBS-T, each wash lasting 10 min. Primary antibodies (1:1000 dilution) were added, and the membrane was incubated in a 4 °C cold room overnight. The membrane was washed three times with TBS-T for 10 min each, shaking at 80 rpm. Secondary antibodies (1:1000 dilution) were then added, and after one and a half hours, the membrane was washed with TBS-T three times for 10 min each, shaking at 80 rpm. Color development reagent was added, and images of the PVDF membrane were captured using a chemical luminescence imaging system (BIOSTEP Celvin^®^) to record the results. Western blot original images can be found in Appendix A.

The antibodies utilized in this study were as follows: anti-phospho p38 (1:1000; Affinity, Cincinnati, OH, USA, #AF4001), anti-p38 (1:1000; Cell Signaling, Danvers, MA, USA, #9212), anti-phospho ERK (1:1000; Cell Signaling, Danvers, MA, USA, #4370), anti-ERK (1:1000; Cell Signaling, Danvers, MA, USA, #9102), anti-phospho JNK (1:1000; Cell Signaling, Danvers, MA, USA, #4668), anti-JNK (1:1000; Cell Signaling, Danvers, MA, USA, #9252), anti-phospho IκB (1:1000; Cell Signaling, Danvers, MA, USA, #9246), anti-β-actin (1:1000; Cell Signaling, Danvers, MA, USA, #4970), anti-phospho NF-κB (1:1000; ABclonal, Woburn, MA, USA, #AP0123), and anti-NF-κB (1:1000; Cell Signaling, Danvers, MA, USA, #8242). Secondary antibodies were as follows, anti-rabbit IgG antibodies (Cell Signaling, Danvers, MA, USA, #7074) and anti-mouse IgG antibodies (Cell Signaling, Danvers, MA, USA, #7076).

### 2.6. In Vivo Experiments

The effect of brianolide on inflammation was observed under the DNCB-induced mouse model of atopic dermatitis. BALB/c mice were used for in vivo experiments and divided into five groups: a control group, a DMSO-only group, a DNCB experimental group, brianolide treatment groups (3 mg/kg with DNCB, 10 mg/kg with DNCB), and a dexamethasone treatment group (0.2 mg/kg with DNCB). There were three mice in each group. Dexamethasone, a corticosteroid with anti-inflammatory, swelling-reducing, and anti-allergic properties used to treat diseases such as arthritis, blood disorders, and hormonal or immune system deficiencies, was included as a positive control. Male BALB/c mice aged eight weeks were selected for this experiment.

Brianolide and dexamethasone were dissolved in DMSO, while DNCB was dissolved in 75% ethanol using ultrasonic agitation. Brianolide (3 mg/kg or 10 mg/kg) and dexamethasone (0.2 mg/kg) were administered via intraperitoneal (IP) injection at a volume of 25 μL. In order to test possible toxicity of DMSO, the mice in the DMSO-only group received 25 μL of DMSO via intraperitoneal injection daily for 7 days. DNCB was applied topically at a volume of 100 μL to the dorsal skin and 20 μL to the right ear.

Three days prior to the start of the experiment (Day-3), the mice were anesthetized, and dorsal hair was removed using a razor and depilatory cream. After three days of rest and acclimatization (until Day 1), the mice were checked for normal health status and the absence of abnormalities in the skin of the depilated area on their backs before proceeding with the experiment.

Baseline physiological data, including TEWL, erythema, ear thickness, and blood flow, were measured on Day 1. Throughout the experiment, photographs documented changes in the appearance of the skin and ears. Transepidermal water loss (TEWL), an indicator of intact skin barrier function, and erythema of the mouse dorsal skin were assessed every day with MPA-580 (Courage & Khazaka, Cologne, Germany). The real-time regional distribution of dorsal skin blood flow was detected using laser speckle contrast imaging (LSCI). To ensure stable environmental conditions for skin parameter evaluation, all assessments were conducted in a temperature- and humidity-controlled chamber.

The experiment consisted of two phases: (Figure 2)

Phase One (Day 1–Day 4): After measuring the baseline values on Day 1, 1% DNCB was evenly applied to the dorsal skin and right ear of the mice to induce initial atopic dermatitis (AD).

Phase Two (Day 5–Day 14): Daily intraperitoneal drug injections commenced on Day 5. To maintain the inflammatory state, a 0.5% DNCB solution was reapplied to the dorsal skin and right ear on Days 8, 11, and 14 in all experimental groups (DNCB alone and treatment groups). Skin physiological parameters were measured, and photographs were taken the day following each DNCB application. Drug treatments were continued daily throughout this phase. The mice were sacrificed on Day 15 after a final measurement.

### 2.7. Data Analysis

The data were analyzed using SigmaPlot software (Version 10.0, Systat Software, Inc., San Jose, CA, USA) and presented as mean ± S.E.M. Statistical analysis was performed using an unpaired two-tailed Student’s *t*-test. A significance level was set at *p* < 0.05, marked with * or #, while *p*-values less than 0.01 were marked with ** or ##.

## 3. Results

### 3.1. Analysis of Brianolide’s Effect on Keratinocyte Viability

HaCaT were treated with various concentrations of brianolide (1–50 µM) for 24 h. Cell viability, assessed using the MTT assay, indicated that brianolide did not produce significant cytotoxic effects across this concentration range, with only a slight inhibition observed at the highest concentration of 50 µM (Figure 3). These findings demonstrate that brianolide maintains high viability in HaCaT cells across a range of concentrations, supporting its potential for further pharmacological studies without adverse effects on cell health. Consequently, a concentration range of 1–10 µM was selected for subsequent pharmacological experiments.

### 3.2. Downregulation of Pro-Inflammatory Cytokine mRNA Expression in HaCaT Following Treatment with Brianolide Under TNF-α/IFN-γ Stimulation

TNF-α and IFN-γ are potent inducers of inflammatory responses in HaCaT cells, leading to increased production of pro-inflammatory cytokines such as IL-1β, IL-6, IL-8, and MDC. These cytokines play significant roles in the pathogenesis of various skin inflammatory conditions. Treatment with brianolide resulted in a significant reduction in the mRNA expression levels of IL-1β, IL-6, and IL-8 in HaCaT cells stimulated by TNF-α and IFN-γ (Figure 4). This suggests that brianolide effectively mitigates the inflammatory response triggered by these cytokines. The anti-inflammatory effects of brianolide may be attributed to its ability to inhibit key signaling pathways involved in inflammation, such as the NF-κB and MAPK pathways, which are crucial for the transcriptional activation of pro-inflammatory cytokines. The findings indicate that brianolide could be a promising therapeutic agent for managing inflammatory skin diseases by reducing the expression of critical pro-inflammatory mediators.

### 3.3. Brianolide Decreased Phosphorylation of Critical MAPK Proteins in TNF-α/IFN-γ-Stimulated HaCaT Cells

The MAPK signaling pathway is closely related to the activation of TNF-α/IFN-γ and regulates various downstream reactions in keratinocytes [29,30,31]. Specifically, the phosphorylation ratios (p-ERK/ERK, p-JNK/JNK, and p-p38/p38) were markedly elevated in the TNF-α/IFN-γ group compared to the control group, indicating the activation of the MAPK pathway during inflammatory stimulation. However, brianolide treatment at concentrations of 1, 3, and 10 μM significantly reduced these phosphorylation levels in a dose-dependent manner. This reduction suggests that brianolide effectively inhibits the activation of the MAPK signaling pathway, which plays a crucial role in mediating inflammatory responses. (Figure 5)

### 3.4. Brianolide Inhibited the Activation of IκB and NF-κB

The canonical NF-κB activation pathway involves phosphorylation of IκB proteins by the IκB kinase (IKK) complex and subsequent degradation [32]. Brianolide appears to interfere with this degradation process, thereby reducing the degradation of IκBα (Figure 6A). The study also showed that Brianolide inhibited the activation of NF-κB, a key transcription factor involved in the inflammatory response (Figure 6B). Therefore, this inhibition is primarily mediated through the regulation of IκB proteins, particularly IκBα. NF-κB remains bound to IκBα in the cytoplasm, preventing its translocation to the nucleus and subsequent transcriptional activation of pro-inflammatory genes.

### 3.5. Brianolide Resulted in a Notable Decrease in Ear Thickness and Dorsal Skin Inflammation in DNCB-Induced Mice

An in vivo animal model of atopic dermatitis was induced in BALB/c mice using 2,4-dinitrochlorobenzene (DNCB). The induction involved an initial application of 1% DNCB on Day 1 to provoke the first phase of inflammation, followed by repeat applications of 0.5% DNCB on Days 8, 11, and 14 to maintain a sustained inflammatory response over a two-week experiment duration. The treatment groups received daily intraperitoneal injections of brianolide (3 mg/kg or 10 mg/kg) dissolved in dimethyl sulfoxide (DMSO) or dexamethasone (0.2 mg/kg, as a positive control) for ten consecutive days, starting on Day 5. The control and DNCB experimental groups were also included. In order to test possible toxicity of DMSO, a group of mice receiving intraperitoneal injection of DMSO for 7 days was also included. Visual assessments and measurements were performed throughout the experiment. Photographic records documented changes in the appearance of the dorsal skin and ears. Ear thickness of the right ear (treated with DNCB) was measured and recorded, with the left ear serving as an untreated control. Baseline physiological values were recorded before induction.

DNCB application successfully induced an atopic dermatitis-like inflammatory response. The DNCB experimental group showed severe redness and swelling on the dorsal skin compared to the control group, with significant scaling and visible wounds. Significant redness and swelling were also observed in the right ear of the DNCB group (Figure 7A). The results revealed that there was no obvious toxicity in the DMSO-only group. Brianolide treatment demonstrated notable effects. Brianolide at 10 mg/kg effectively alleviated symptoms of epidermal hyperplasia and redness. Continuous observation of the dorsal skin in treatment groups showed a noticeable trend toward reduced severity, although some redness persisted compared to the control. The treated groups also exhibited reduced ear swelling compared to the DNCB group (Figure 7B).

Observations of ear conditions indicated a significant trend toward relief in treated groups compared to the DNCB experimental group. These experimental results from visual assessments of changes in the dorsal skin and ears confirm that brianolide effectively inhibits the redness and inflammatory response induced by DNCB. In this DNCB-induced AD model, treatment with brianolide resulted in a notable decrease in ear thickness and dorsal skin inflammation. This reduction highlights brianolide’s anti-inflammatory properties and suggests its effectiveness in alleviating inflammatory responses associated with skin conditions like atopic dermatitis.

### 3.6. Brianolide’s Direct Effects on Blood Flow

Brianolide’s effects on skin blood flow were assessed using laser speckle contrast imaging (LSCI). By modulating inflammatory pathways such as MAPK and NF-κB, brianolide may indirectly influence blood flow by reducing inflammation-related vascular responses. Inflammation often leads to increased blood flow due to the release of pro-inflammatory mediators and vasodilation; thus, reducing inflammation could normalize blood flow dynamics in affected tissues. In the DNCB-induced model, the DNCB-alone group displayed higher blood flow compared to the control. However, groups pretreated with brianolide (3 mg/kg or 10 mg/kg) before DNCB application displayed significantly reduced blood flow (Figure 8). These results suggest that brianolide may play a role in normalizing skin blood flow, possibly through reducing inflammation mediated vasodilation.

### 3.7. Brianolide Improves Physiological Parameters Such as TEWL, Hydration, and Erythema

Atopic dermatitis is characterized by clinical features such as skin lesions, redness (erythema), and increased thickness due to inflammation. It often results in impaired skin barrier function, leading to increased TEWL and reduced hydration levels. These parameters were investigated in DNCB-induced mice using a multifunctional skin physiological testing system. Baseline physiological values for TEWL, erythema, and ear thickness were recorded at the beginning of the experiment.

Following the initial DNCB induction, TEWL and erythema values showed a significant increase by Day 9 in the DNCB experimental group, indicating a marked trend of elevated physiological parameters and suggesting the barrier function of the mice’s skin was compromised. Measurements of ear thickness also revealed a notable increase in the right ear thickness of mice in the DNCB group.

In contrast, groups treated with brianolide at doses of 3 mg/kg and 10 mg/kg, as well as dexamethasone at 0.2 mg/kg, showed improvements in TEWL, erythema, and ear thickness, with these physiological values being superior to those in the DNCB group (Figure 9A–C). Skin hydration levels, also measured, showed improvement with brianolide treatment (Figure 9D). This indicates that brianolide treatment helps restore damaged skin barrier function and regulate abnormal vascular responses.

## 4. Discussion

Marine natural products have become important sources for drug development due to their unique chemical structures and biological activities. These compounds originate from marine organisms such as sponges, corals, and algae, which produce a variety of secondary metabolites as part of their defense mechanisms. Potential applications include anticancer drugs such as halichondrin [33] and cytarabine [34]. Studies have shown that briarane-type diterpenoid compounds exhibit beneficial effects on the skin and possess anti-inflammatory properties. For example, it can reduce TPA-induced skin inflammation in mice [23], or it can reduce the inflammatory response caused by LPS-induced RAW 264.7 cells, and reduce skin damage caused by DNCB [25]. This study first utilized TNF-α/IFN-γ-induced human keratinocyte cell lines (HaCaT) to simulate the inflammatory response of atopic dermatitis. The results showed that brianolide inhibited the expression of cytokines and chemokines through modulating the MAPK and NF-κB pathways. These in vitro results provided preliminary evidence for the potential mechanism of brianolide in inhibiting atopic dermatitis-related inflammation at the cellular level, particularly by regulating key inflammatory signaling pathways. Subsequently, this study further utilized a DNCB-induced atopic dermatitis animal model to demonstrate that brianolide has anti-inflammatory effects on the DNCB-induced inflammatory response. Specifically, brianolide treatment significantly reduced TEWL, ear thickness, erythema, and epidermal blood flow. These in vivo experimental results echoed the findings of the in vitro cellular experiments, collectively supporting the anti-inflammatory effects of brianolide as a potential therapeutic agent for atopic dermatitis.

To further understand the anti-inflammatory mechanism of brianolide, this study explored its effects on key inflammatory mediators and signaling pathways. Atopic dermatitis is characterized by the overproduction of pro-inflammatory cytokines (such as IL-1β, IL-6, IL-8) [35,36,37,38,39]. This study found that brianolide could significantly reduce the mRNA expression levels of IL-1β, IL-6, and IL-8 in TNF-α/IFN-γ-stimulated HaCaT cells. These cytokines play a crucial role in the pathogenesis of atopic dermatitis; for example, increased IL-1β secretion following the activation of the MAPK and NF-κB pathways promotes the production of additional pro-inflammatory cytokines [35,37]. IL-8 mainly mediates neutrophil chemotaxis [35,38]. The inhibitory effect of brianolide on the expression of these cytokines suggests its potential to alleviate the inflammatory response in atopic dermatitis (Figure 4). The MAPK pathways (including ERK, JNK, and p38) play an important role in the inflammatory response. The in vitro experiments in this study demonstrated that brianolide treatment reduced the phosphorylation levels of key MAPK proteins in TNF-α/IFN-γ-stimulated HaCaT cells (Figure 5). This indicates that brianolide can effectively inhibit the activation of the MAPK pathway, thereby reducing inflammatory signal transduction. NF-κB is a key transcription factor responsible for regulating the expression of multiple pro-inflammatory genes. In atopic dermatitis, the activation of NF-κB increases the production of pro-inflammatory cytokines and affects the proliferation and differentiation of keratinocytes [40,41,42]. This study showed that brianolide inhibited the activation of IκB and the nuclear translocation of NF-κB, which is achieved by reducing the degradation of IκBα, keeping NF-κB bound to IκBα in the cytoplasm and preventing it from entering the nucleus and activating the transcription of pro-inflammatory genes (Figure 6). Therefore, brianolide may interrupt the vicious cycle of inflammation and impaired skin barrier in atopic dermatitis by inhibiting the activation of NF-κB.

Patients with atopic dermatitis often have an impaired skin barrier function, leading to increased TEWL and dry skin. The inflammatory response can also cause vasodilation, leading to erythema and increased blood flow [43,44]. The animal experimental results of this study showed that brianolide treatment improved TEWL, hydration, erythema, and ear thickness in DNCB-induced mice (Figure 9). This indicates that brianolide not only alleviates the inflammatory response but also helps restore the damaged skin barrier function and regulate abnormal vascular responses [45,46]. Notably, the inhibitory effect of brianolide on skin blood flow is consistent with the observed reduction in erythema (Figure 7 and Figure 8). Although this study noted improvements in TEWL and skin hydration, further research is needed to determine whether skin barrier-related proteins, such as filaggrin and ceramides, have recovered.

Overall, the in vitro and in vivo experimental results of this study consistently demonstrate that brianolide, a compound isolated from the soft coral *Briareum stechei*, effectively inhibits atopic dermatitis-related inflammation by regulating the MAPK and NF-κB signaling pathways. This study is the first to use a TNF-α/IFN-γ-induced human keratinocyte cell line to model the inflammatory response of atopic dermatitis and to elaborate on the mechanism of action of brianolide in detail. These findings provide new insights for developing brianolide as a therapeutic or adjuvant drug for atopic dermatitis, with the potential to reduce reliance on steroids or immunosuppressants.

## 5. Conclusions

This study investigated brianolide, a briarane compound, revealing its potential as an anti-inflammatory agent in atopic dermatitis by modulating MAPK and NF-κB pathways. Future research should focus on clinical translation, including optimized formulations and safety trials, mechanistic studies to pinpoint molecular targets, exploration of its use as a steroid-sparing agent in other skin conditions, and development of derivatives with improved properties. Brianolide shows promise as a therapeutic option, potentially reducing reliance on corticosteroids while effectively managing inflammation.

## Figures and Tables

**Figure 1 biomolecules-15-00871-f001:**
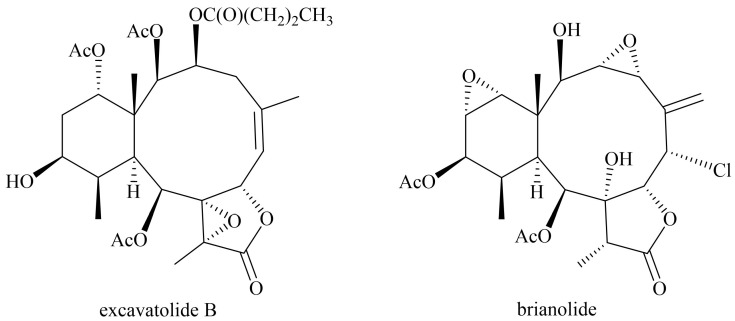
The structure of ecavatolide B and brianolide.

**Figure 2 biomolecules-15-00871-f002:**
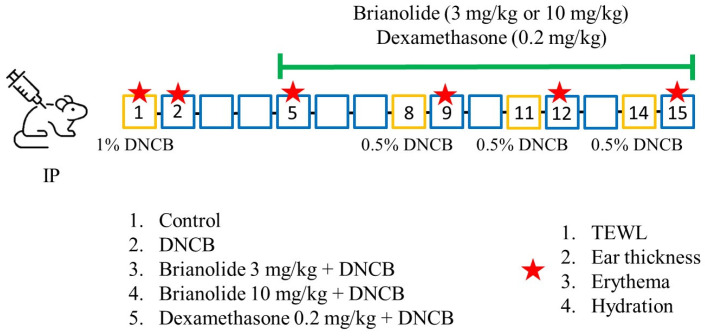
Experimental timeline summary. Shaving the fur three days before the experiment; D1: Baseline physiological measurements, administration of 1% DNCB; D2: Baseline physiological measurements; D5–D14: Daily administration of brianolide or dexamethasone; D8, D11, D14: 0.5% DNCB reapplication; D9, D12: Baseline physiological measurements; D15: Physiological measurements and sacrifice of mice.

**Figure 3 biomolecules-15-00871-f003:**
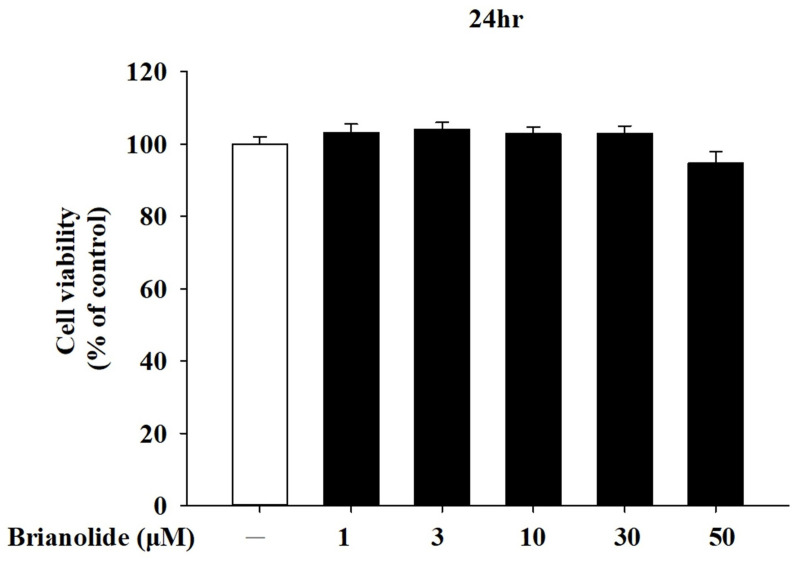
Cell viability in human keratinocytes (HaCaT) after treatment with different concen trations of brianolide using the MTT Assay.

**Figure 4 biomolecules-15-00871-f004:**
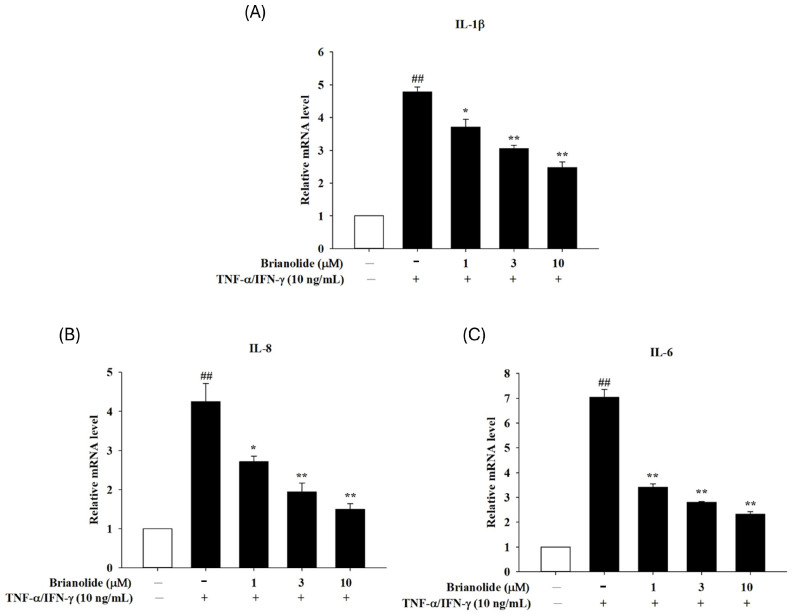
Changes in mRNA expression levels of (**A**) IL-1β, (**B**) IL-8, and (**C**) IL-6 under the action of brianolide. The cells were pretreated with different concentrations of brianolide (1, 3, and 10 μM) for 1 h, followed by treatment with TNF-α/IFN-γ (10 ng/mL) for 1 h. Total RNA was isolated, and the mRNA expression levels were determined using qPCR. Values represent the mean ± SEM from three independent experiments. ## *p* < 0.01 compared with the no-treatment condition; * *p* < 0.05 and ** *p* < 0.01 compared with the TNF-α/IFN-γ treatment condition.

**Figure 5 biomolecules-15-00871-f005:**
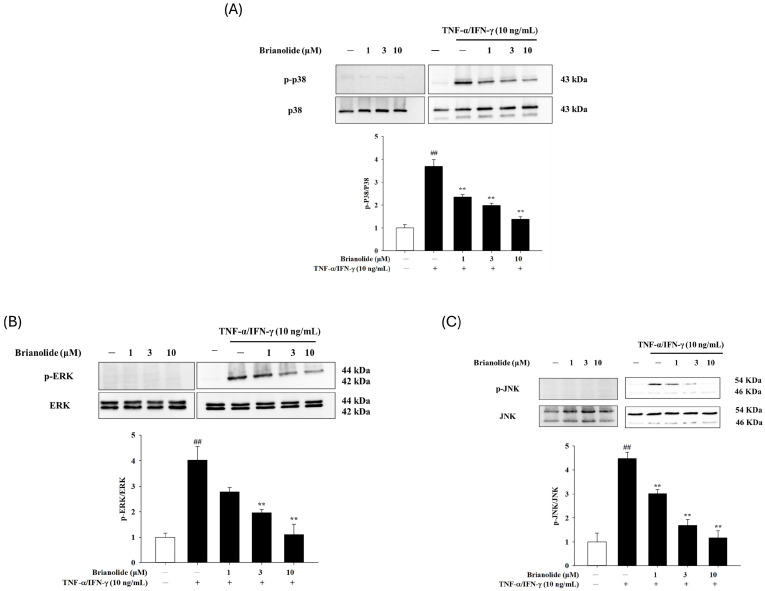
Brianolide reduced TNF-α/IFN-γ (10 ng/mL)-stimulated (**A**) p38, (**B**) ERK, and (**C**) JNK phosphorylation in HaCaT cells. The cells were pretreated with brianolide (1, 3, and 10 μM) for 1 h, followed by stimulation with TNF-α/IFN-γ (10 ng/mL) for 0.5 h. Western blots were analyzed quantitatively. The values represent the mean ± SEM from the three independent experiments. ## *p* < 0.01 compared with the control group; ** *p* < 0.01 compared with the TNF-α/IFN-γ treatment group.

**Figure 6 biomolecules-15-00871-f006:**
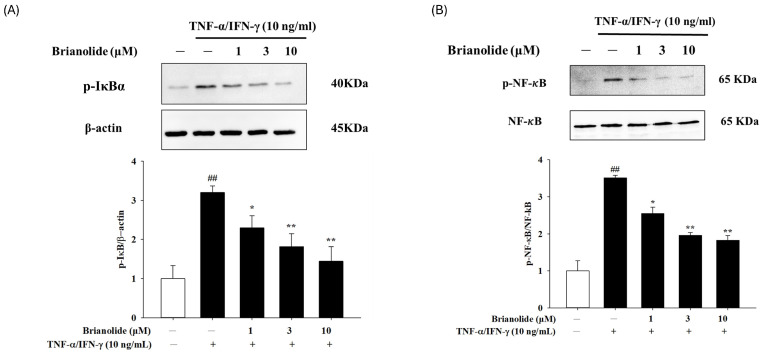
Brianolide reduced TNF-α/IFN-γ (10 ng/mL) stimulated (**A**) IκBα, (**B**) NF-κB activation in HaCaT cells. The cells were pretreated with brianolide (1, 3, and 10 μM) for 1 h, followed by stimulation with TNF-α/IFN-γ (10 ng/mL) for 1 h. Western blots were analyzed quantitatively. Values represent the mean ± SEM from the three independent experiments. ## *p* < 0.01 compared with the control group; * *p* < 0.05 and ** *p* < 0.01 compared with the TNF-α/IFN-γ treatment group.

**Figure 7 biomolecules-15-00871-f007:**
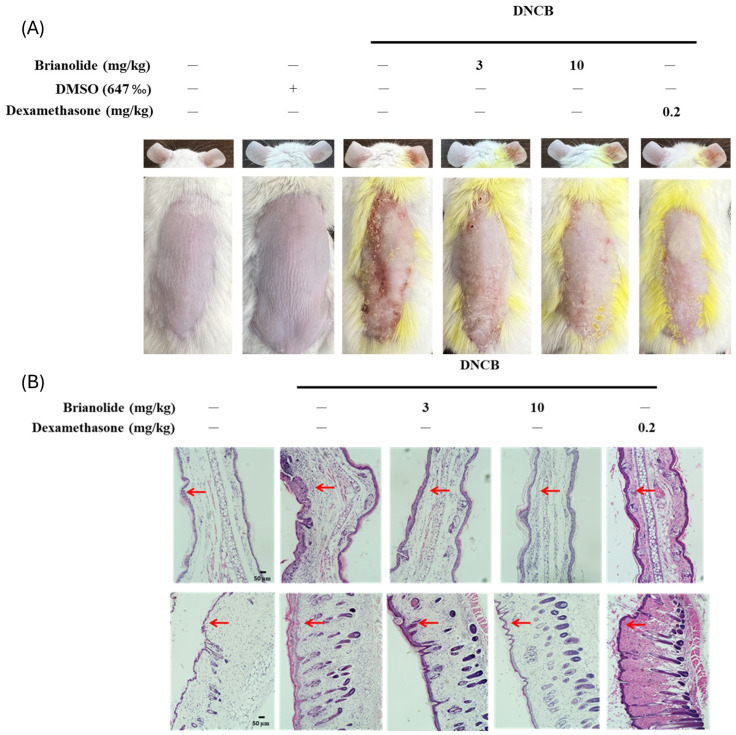
Phenotypic and histological analysis to assess the effect of brianolide on DNCB-induced atopic dermatitis. (**A**) Phenotypic presentation of mouse skin after 15 days of treatment; (**B**) Histological examination of ear tissue after H&E staining from each treatment group (upper panel: ×100 magnification; lower panel: ×400 magnification). The red arrow indicated the epidermal layer of skin. Application of DNCB increased the epidermal thickness, while treatment with brianolide reduced the thickness.

**Figure 8 biomolecules-15-00871-f008:**
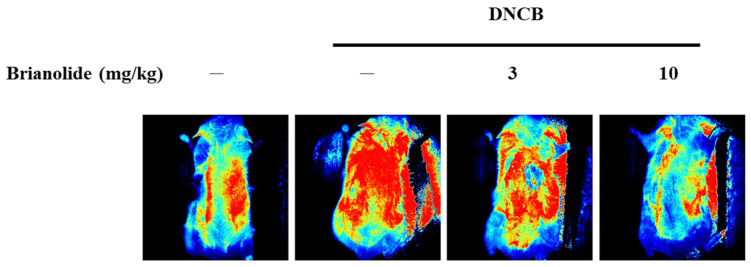
Effect of Brianolide on DNCB-induced skin blood flow in mice assessed by Laser Speckle Contrast Imaging. DNCB application significantly increased cutaneous blood flow. Treatment with brianolide at both 3 mg/kg and 10 mg/kg markedly attenuated this DNCB-induced elevation in blood flow. Warm colors (red/yellow) depict higher blood flow, whereas cool colors (blue) represent lower blood flow.

**Figure 9 biomolecules-15-00871-f009:**
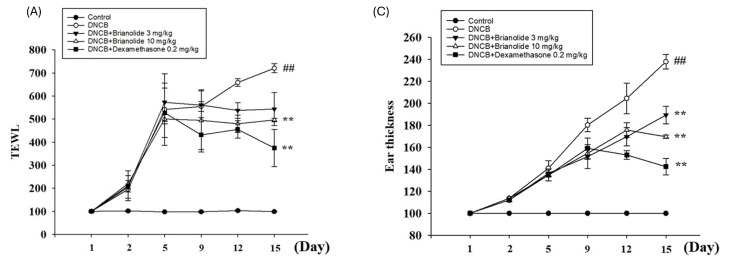
Effects of brianolide on physiological parameters, including (**A**) transepidermal water loss, (**B**) erythema, (**C**) ear thickness, and (**D**) hydration, in DNCB-induced BALB/c mouse. Values represent the mean ± SEM from at least three independent experiments. ## *p* < 0.01 compared with the control group; ** *p* < 0.01 compared with the DNCB-induced group.

**Table 1 biomolecules-15-00871-t001:** Human primer sequences for RT-qPCR.

Genes	Primers	Sequence(5′-3′)
IL-1β	Forward	CTC TCA CCT CTC CTA CTC ACT
	Reverse	ATC AGA ATG TGG GAG CGA AT
IL-6	Forward	CGA GCC CAC CGG GAA CGA AA
	Reverse	GGA CCG AAG GCG CTT GTG GAG
IL-8	Forward	ACT GAG AGT GAT TGA GAG TGG AC
	Reverse	AAC CCT CTG CAC CCA GTT TTC

## Data Availability

The data presented in this study is available in the article/Appendix A.

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
