# Peer review of "Brianolide from Briareum stechei Attenuates Atopic Dermatitis-like Skin Lesions by Regulating the NFκB and MAPK Pathways"

_biomolecules, 2025, doi:10.3390/biom15060871_

Round 1
Reviewer 1 Report
Comments and Suggestions for Authors
Work is good and well presented
There are minor corrections as given below:
- in vitro and in vivo should be always in italics.
- Which instrument was used for TEWL measurement ?
- Enclose animal ethical committee approval letter
- Figure 5 and 6 needs more detail caption explaining it
- Which instrument was used to measure erythema and hydration?
- In Conclusion future prospects need to be added.
Author Response
Work is good and well presented
There are minor corrections as given below:
- in vitro and in vivo should be always in italics.
Reply: Thank you for your insightful comment. We have carefully reviewed the manuscript and ensured that in vitro and in vivo are consistently italicized throughout the text in accordance with standard academic conventions.
- Which instrument was used for TEWL measurement ?
Reply: Thank you for your valuable feedback. Transepidermal water loss (TEWL), serving as an indicator of intact skin barrier function, as well as erythema of the dorsal skin in mice, were evaluated daily using the MPA-580 instrument (Courage & Khazaka, Cologne, Germany) prior to and following the specified treatments. This information has been incorporated into the relevant sections of the manuscript for clarity and comprehensiveness.
- Enclose animal ethical committee approval letter.
Reply: Thank you for your comment. The animal ethical committee approval letter has been included in the supplementary section of this response-to-reviewer letter for your reference.
- Figure 5 and 6 needs more detail caption explaining it
Reply: Thank you for your feedback. We have revised and provided more detailed captions for Figures 5 and 6 to enhance their clarity and explanatory value.
- Which instrument was used to measure erythema and hydration?
Reply: Thank you for your insightful question. Erythema and transepidermal water loss (TEWL), an indicator of skin hydration and barrier function, were assessed daily using the MPA-580 instrument (Courage & Khazaka, Cologne, Germany) before and after the specified treatments. This information has been included in the relevant sections of the manuscript for clarity and completeness.
- In Conclusion future prospects need to be added.
Reply: Thank you for your valuable suggestion. We have revised the conclusion to include future prospects, providing a more comprehensive outlook.
(Supplementary material)

Reviewer 2 Report
Comments and Suggestions for Authors
Unfortunately, the manuscript does not have line numbering. This makes writing a review very difficult. I have made notes in red font on the left-hand side of the article file.
The article is generally written very clearly and the introduction describes the scientific problem well. However, some of the methods concerning the determination of phosphorylated proteins are stated very briefly, providing insufficient detail to understand the obtained results. These results are also not described in sufficient detail, which calls into question the conclusions that the authors draw about the regulation of MAPK and NF-κB signalling pathways by brianolide. There is also a lack of description of the brainolide substance itself: how it was obtained and whether it is sufficiently homogeneous. In the animal experiments, the number of individuals in each group is not specified, nor is the statistical treatment of the results.

Author Response
- Reference 18 does not match the information given
Reply: Thank you for your insightful feedback and we have made thorough check on the citation to ensure the correctness and clarity.
- (Figure2) The title of the drawing should be changed. It should reflect the process, method or structure, and should not duplicate the statement of results in the article's text. The second column is unclear, as is the concentration of brianolide. The control group should be explained.
Reply: We sincerely appreciate your feedback. In response, we have revised the figure caption to more accurately reflect the process and methodology described. Additionally, the figure itself has been updated to ensure correctness and clarity for each group. We believe these changes improve the overall quality and accuracy of the presentation. Thank you for bringing this to our attention.
- It is necessary to explain why the conclusion of inhibition was drawn and provide the relevant references.
Reply: Thank you for your valuable feedback. We have revised this section to enhance its clarity and ensure better comprehension.
- (Figure 3) The title of the drawing should be changed. It should reflect the process, method or structure, and should not duplicate the statement of results in the article's text. The caption for the drawing is unfortunate in that it does not explain anything. What are p-ERK and ERK (also in 3A and 3C), and what method was used to analyse them? How were the values in the diagrams' columns obtained? If the amount of phosphorylated proteins is decreasing, why isn't the amount of nonphosphorylated proteins changing? And why is there such a difference between pp38 and p38 (also in 3B), when TNF and IFN aren't added?How would you explain that?
Reply: We have revised the caption to more effectively illustrate the process and results of the experiment. The Y-axis represents the relative fold change of phosphorylated proteins and total proteins. The figure demonstrates that, upon INF-γ/TNF-α exposure, the levels of phosphorylated p-38, ERK, and JNK increased, indicating the activation of the MAPK pathways. Conversely, exposure to brianolide resulted in a dose-dependent decrease in the phosphorylation of MAPK-related proteins. Detailed methods of how to process the western blot images were described in the method section.
- (Figure 4) The title of the drawing should be changed. It should reflect the process, method or structure, and should not duplicate the statement of results in the article's text. The process by which such conclusions are drawn from the presented image should be better explained. Why was β-actin measured, and how does it affect IκB degradation? How can a decrease in NF-κB phosphorylation indicate inhibition of NF-κB activation? Why is the phosphorylated form of NF-κB reduced, yet the normal form is not?
Reply:
- We have revised the caption to better reflect the process.
- Why was β-actin measured, and how does it affect IκB degradation?β-actin was measured as a loading control to ensure equal protein loading across all samples during Western blot analysis. It serves as a reference to normalize the expression levels of the target proteins, such as IκB and NF-κB, allowing for accurate comparison between experimental groups.
β-actin itself does not directly affect IκB degradation. The degradation of IκB is regulated by the ubiquitin-proteasome pathway, which is activated following phosphorylation of IκB by IκB kinase (IKK) [1]. The inclusion of β-actin as a control ensures that observed changes in IκB levels are due to experimental treatments rather than variability in sample loading or transfer efficiency.
- How can a decrease in NF-κB phosphorylation indicate inhibition of NF-κB activation?
Phosphorylation of NF-κB is a critical step in its activation process. Upon stimulation (e.g., by TNF-α/IFN-γ), upstream kinases such as IKK phosphorylate IκB, leading to its degradation and the subsequent release of NF-κB. Once free, NF-κB is translocated freely to nucleus to initiate further response. P65, subunit of NF-κB, may undergo phosphorylation, which enhances its transcriptional activity and facilitates its nuclear translocation to activate target gene expression.[2–5] Moreover, the P50 subunit of the canonical NF-κB pathway can be phosphorylated. Phosphorylation of P50 at Serine 328 has been shown to regulate NF-κB transcriptional activity by influencing its DNA-binding ability[4]. Additionally, Hou et al. demonstrated that phosphorylation of P50 at Serine 337 is essential for increasing its DNA-binding affinity. For the P65 subunit, phosphorylation at Serine 529 and Serine 536 by casein kinase II and IKKα, respectively, enhances NF-κB's transactivation potential [6].
- Why is the phosphorylated form of NF-κB reduced, yet the normal form is not?
The total (unphosphorylated) form of NF-κB represents the basal level of the protein, which is typically stable under normal conditions. Experimental treatments, such as brianolide, may specifically target the signaling pathways that regulate phosphorylation without affecting the overall synthesis or degradation of NF-κB.
As a result, the phosphorylated form decreases due to inhibition of the upstream kinases responsible for its phosphorylation, while the total protein level remains unchanged. This distinction highlights the regulatory role of phosphorylation that modulates NF-κB activity independently of its overall abundance.
- The wording needs to be more precise. These results only indicate a decrease in blood flow, which could have different causes, and brianolide may not affect inflammation.
Reply: We have made changes in this section for better accuracy and clarity.
- Do you need to explain how TEWL, hydration and erythema were measured? There's no mention of this in the methods!
Reply: Transepidermal water loss (TEWL), an indicator of intact skin barrier function, and er-ythema of the mouse dorsal skin were assessed every day with MPA-580 (Courage & Khazaka, Cologne, Germany). This information was added in the method section.
- (Figure 6) The title of the drawing should be changed. It should reflect the process, method or structure, and should not duplicate the statement of results in the article's text.
Reply: We have modified the captions. Thank you for your valuable feedback.
- (Figure 7) The X-axis is not labelled. The title of the drawing should be changed. It should reflect the process, method or structure, and should not duplicate the statement of results in the article's text.
Reply: We have modified the captions and figures as you suggested. Thank you for your feedback.
- [9] - the reference does not correspond to the information given in the article. It is necessary to specify how the brianolide was obtained and purified
Reply: The references have been thoroughly reviewed, and necessary modifications have been made accordingly. The brianolide was extracted and purified as previously published work by Prof. Ping-Jyun Sung. [7]
- What was the control value when converted to a viability percentage? If the brianolide was in DMSO, was this the control?
Reply: The experiment aimed to evaluate the cytotoxic effects of brianolide on HaCaT cells using the MTT assay. The control group, consisting of cells not exposed to brianolide, was used as a baseline for comparison, with cell viability expressed as a percentage relative to this group. The MTT assay involved adding MTT reagent to both treated and untreated cells, where metabolically active cells reduced MTT to formazan crystals. DMSO was then used to dissolve these crystals uniformly, ensuring accurate absorbance measurements. This consistent procedure across all groups allowed for the assessment of brianolide's impact on cell viability by comparing the absorbance values of treated cells to the control group.
- Since method statements are often provided when reporting results, it is not immediately clear what was actually measured by the Western blot. While all the standard steps of the method are described in detail, the specific requirements for using Western blotting in these experiments are not specified. Which primary and secondary antibodies were used in all cases? How were the phosphorylated and unphosphorylated forms of the proteins visualised?
Reply: The antibodies used in this study were as follows: anti-phospho p38 (1:1000; Affinity, USA #AF4001), anti-p38 (1:1000; Cell Signaling, USA #9212), anti-phospho ERK (1:1000; Cell Signaling, USA #4370), anti-ERK (1:1000; Cell Signaling, USA #9102), anti-phospho JNK (1:1000; Cell Signaling, USA #4668), anti-JNK (1:1000; Cell Signaling, USA #9252), anti-phospho IκB (1:1000; Cell Signaling, USA #9246), anti-β-actin (1:1000; Cell Signaling, USA #4970), anti-phospho NF-κB (1:1000; ABclonal, USA #AP0123), and anti-NF-κB (1:1000; Cell Signaling, USA #8242). These primary antibodies were incubated overnight at 4°C. Secondary antibodies included anti-rabbit IgG (1:2000, 1:5000, 1:10000; Cell Signaling, USA #7074) and anti-mouse IgG (1:5000; Cell Signaling, USA #7076), which were incubated at room temperature for 90 minutes. Images of the PVDF membrane were captured using a chemical luminescence imaging system (BIOSTEP Celvin®) to record the results.
- How many animals were in each group? How was the statistical analysis performed? Where in the article is this information given?
Reply: Thank you for your insightful feedback. There were 3 mice in each group. The statistical analysis was presented in the data analysis section. We have revised the article to enhance clarity.
- It is necessary to specify the method used to measure the physiological parameters
Reply: Thank you for your insightful feedback. Transepidermal water loss (TEWL), an indicator of intact skin barrier function, and erythema of the mouse dorsal skin were assessed every day with MPA-580 (Courage & Khazaka, Cologne, Germany). This information was added in the method section to enhance clarity.
References:
- Kanarek, N.; Ben-Neriah, Y. Regulation of NF-κB by Ubiquitination and Degradation of the IκBs. Immunol Rev 2012, 246, 77–94, doi:10.1111/j.1600-065X.2012.01098.x.
- Begalli, F.; Bennett, J.; Capece, D.; Verzella, D.; D’Andrea, D.; Tornatore, L.; Franzoso, G. Unlocking the NF-κB Conundrum: Embracing Complexity to Achieve Specificity. Biomedicines 2017, 5, 50, doi:10.3390/biomedicines5030050.
- Zhong, H.; Voll, R.E.; Ghosh, S. Phosphorylation of NF-Kappa B P65 by PKA Stimulates Transcriptional Activity by Promoting a Novel Bivalent Interaction with the Coactivator CBP/P300. Mol Cell 1998, 1, 661–671, doi:10.1016/s1097-2765(00)80066-0.
- Christian, F.; Smith, E.L.; Carmody, R.J. The Regulation of NF-κB Subunits by Phosphorylation. Cells 2016, 5, 12, doi:10.3390/cells5010012.
- Zhong, H.; May, M.J.; Jimi, E.; Ghosh, S. The Phosphorylation Status of Nuclear NF-ΚB Determines Its Association with CBP/P300 or HDAC-1. Molecular Cell 2002, 9, 625–636, doi:10.1016/S1097-2765(02)00477-X.
- Hou, S.; Guan, H.; Ricciardi, R.P. Phosphorylation of Serine 337 of NF-κB P50 Is Critical for DNA Binding *. Journal of Biological Chemistry 2003, 278, 45994–45998, doi:10.1074/jbc.M307971200.
- Chen, Y.-Y.; Zhang, Y.-L.; Lee, G.-H.; Tsou, L.K.; Zhang, M.M.; Hsieh, H.-P.; Chen, J.-J.; Ko, C.-Y.; Wen, Z.-H.; Sung, P.-J. Briarenols W–Z: Chlorine-Containing Polyoxygenated Briaranes from Octocoral Briareum Stechei (Kükenthal, 1908). Marine Drugs 2021, 19, 77, doi:10.3390/md19020077.
Reviewer 3 Report
Comments and Suggestions for Authors
This study investigated the anti-atopic efficacy of brianolide. However, the research method seems to need to be modified for the following reasons.
Major coment -
1) In in vitro experiments, while TNFa and IFNg belong to Th2 cytokines, the major cytokines that induce atopy are Th2 cytokines such as IL-4, IL-5, IL-13, and TSLP. Therefore, in vitro anti-inflammatory experiments require Th2 cytokines as stimuli to infer anti-inflammatory effects on atopy. However, in this study, a Th1 cytokine mixture was used, so it is difficult to assume that this is in vitro atopy model.
2) In animal experiments, Brianolide and dexamethasone were dissolved in DMSO and treated via IP injection. However, since DMSO is a vehicle that can exhibit toxicity, a DMSO-only IP treatment group is necessary to see the effect of DMSO, but it is omitted.
Minor coment -
1) Fig. 2) In the second group in the graph, the cytokines mixture only treatment group, brianolide (+) should be changed to (-).
2) Fig. 3) In the JNK experiment, the control group was treated differently from ERK or p38. . An explanation is needed as to why they were handled differently.
3) Fig. 4) The target name of the membrane should be changed from P-IkB to p-IkBa.
Author Response
This study investigated the anti-atopic efficacy of brianolide. However, the research method seems to need to be modified for the following reasons.
Major coment -
- In in vitro experiments, while TNFa and IFNg belong to Th2 cytokines, the major cytokines that induce atopy are Th2 cytokines such as IL-4, IL-5, IL-13, and TSLP. Therefore, in vitro anti-inflammatory experiments require Th2 cytokines as stimuli to infer anti-inflammatory effects on atopy. However, in this study, a Th1 cytokine mixture was used, so it is difficult to assume that this is in vitro atopy model.
Reply: We appreciate the reviewer for raising this significant question about our selection of inflammatory stimuli in the in vitro model. To clarify, the rationale behind using TNF-α and IFN-γ in our HaCaT keratinocyte model stems from their critical roles in representing distinct phases and aspects of atopic dermatitis. While we recognize the pivotal involvement of Th2 cytokines (IL-4, IL-5, IL-13, TSLP) in the pathogenesis of atopic dermatitis, TNF-α and IFN-γ are equally significant inflammatory mediators. Numerous studies have demonstrated that IFN-γ and TNF-α activate epidermal keratinocytes, triggering various signal transduction pathways and contributing to the amplification of inflammation.[1–3] Hung et al. previously demonstrated that TNF-α and IFN-γ stimulate human keratinocytes to induce the expression of large amounts of inflammatory factors (IL-1β, IL-6, IL-8, TSLP, TARC, MDC, and RANTES). [4,5] Therefore, IFN-γ/TNF-α stimulation is usually used as an induction method for in vitro anti-inflammatory skin experiments.[6,7]
- In animal experiments, Brianolide and dexamethasone were dissolved in DMSO and treated via IP injection. However, since DMSO is a vehicle that can exhibit toxicity, a DMSO-only IP treatment group is necessary to see the effect of DMSO, but it is omitted.
Reply: Thank you for your valuable suggestion. We conducted an experiment to evaluate the potential toxicity of DMSO. DMSO was administered via intraperitoneal injection for 7 consecutive days, and no signs of toxicity were observed, as shown in the updated Figure 5. The results revealed that there was no obvious toxicity in DMSO treated mice. Additionally, we have made the corresponding revisions in the manuscript.
Minor coment -
- 2) In the second group in the graph, the cytokines mixture only treatment group, brianolide (+) should be changed to (-).
Reply: We have made the revisions to Figure 2. Thank you for your insightful feedback.
- 3) In the JNK experiment, the control group was treated differently from ERK or p38. . An explanation is needed as to why they were handled differently.
Reply: The control group was treated in the same manner as the ERK or P38 groups. We have updated the figure accordingly. Thank you for your valuable feedback.
3) Fig. 4) The target name of the membrane should be changed from P-IkB to p-IkBa.
Reply: We have modified the name of membrane. Thank you for your feedback.
References:
- Gottlieb, A.B.; Chamian, F.; Masud, S.; Cardinale, I.; Abello, M.V.; Lowes, M.A.; Chen, F.; Magliocco, M.; Krueger, J.G. TNF Inhibition Rapidly Down-Regulates Multiple Proinflammatory Pathways in Psoriasis Plaques1. The Journal of Immunology 2005, 175, 2721–2729, doi:10.4049/jimmunol.175.4.2721.
- Mehta, N.N.; Teague, H.L.; Swindell, W.R.; Baumer, Y.; Ward, N.L.; Xing, X.; Baugous, B.; Johnston, A.; Joshi, A.A.; Silverman, J.; et al. IFN-γ and TNF-α Synergism May Provide a Link between Psoriasis and Inflammatory Atherogenesis. Sci Rep 2017, 7, 13831, doi:10.1038/s41598-017-14365-1.
- Kong, L.; Liu, J.; Wang, J.; Luo, Q.; Zhang, H.; Liu, B.; Xu, F.; Pang, Q.; Liu, Y.; Dong, J. Icariin Inhibits TNF-α/IFN-γ Induced Inflammatory Response via Inhibition of the Substance P and P38-MAPK Signaling Pathway in Human Keratinocytes. International Immunopharmacology 2015, 29, 401–407, doi:10.1016/j.intimp.2015.10.023.
- Yang, C.-C.; Hung, Y.-L.; Ko, W.-C.; Tsai, Y.-J.; Chang, J.-F.; Liang, C.-W.; Chang, D.-C.; Hung, C.-F. Effect of Neferine on DNCB-Induced Atopic Dermatitis in HaCaT Cells and BALB/c Mice. International Journal of Molecular Sciences 2021, 22, 8237, doi:10.3390/ijms22158237.
- Lee, K.-S.; Chun, S.-Y.; Lee, M.-G.; Kim, S.; Jang, T.-J.; Nam, K.-S. The Prevention of TNF-α/IFN-γ Mixture-Induced Inflammation in Human Keratinocyte and Atopic Dermatitis-like Skin Lesions in Nc/Nga Mice by Mineral-Balanced Deep Sea Water. Biomedicine & Pharmacotherapy 2018, 97, 1331–1340, doi:10.1016/j.biopha.2017.11.056.
- Su, Y.; Han, Y.; Choi, H.S.; Lee, G.-Y.; Cho, H.W.; Choi, H.; Jang, Y.-S.; Choi, J.H.; Seo, J.-W. Lipid Mediators Derived from DHA Alleviate DNCB-Induced Atopic Dermatitis and Improve the Gut Microbiome in BALB/c Mice. International Immunopharmacology 2023, 124, 110900, doi:10.1016/j.intimp.2023.110900.
- Kim, H.J.; Song, H.-K.; Park, S.H.; Jang, S.; Park, K.-S.; Song, K.H.; Lee, S.K.; Kim, T. Terminalia Chebula Retz. Extract Ameliorates the Symptoms of Atopic Dermatitis by Regulating Anti-Inflammatory Factors in Vivo and Suppressing STAT1/3 and NF-ĸB Signaling in Vitro. Phytomedicine 2022, 104, 154318, doi:10.1016/j.phymed.2022.154318.
Round 2
Reviewer 2 Report
Comments and Suggestions for Authors
There is one general comment remaining on the captioning of the drawings. It seems that I was not clear in the first review. The caption should provide a general overview of the information presented in the figure. Figure 1 has the only correct caption in the article. Below is an example of the caption that should accompany Figure 2: Changes in mRNA expression levels of (A) IL-1β, (B) IL-8 and (C) IL-6 under the action of brianolide'. Cells were pretreated with different concentrations of brianolide (1, 3, and 10 μM) for 1 hour, followed by treatment with TNF-α/IFN-γ (10 ng/mL) for 1 hour. Total RNA was isolated, and the mRNA expression levels were determined using qPCR. Values represent the mean ± SEM from three independent experiments. ##p < 0.01 compared with the no-treatment condition; *p < 0.05 and **p < 0.01 compared with the TNF-α/IFN-γ treatment condition.
All other figures should be labelled similarly.
All other corrections are completely satisfactory to me.
Author Response
There is one general comment remaining on the captioning of the drawings. It seems that I was not clear in the first review. The caption should provide a general overview of the information presented in the figure. Figure 1 has the only correct caption in the article. Below is an example of the caption that should accompany Figure 2: Changes in mRNA expression levels of (A) IL-1β, (B) IL-8 and (C) IL-6 under the action of brianolide'. Cells were pretreated with different concentrations of brianolide (1, 3, and 10 μM) for 1 hour, followed by treatment with TNF-α/IFN-γ (10 ng/mL) for 1 hour. Total RNA was isolated, and the mRNA expression levels were determined using qPCR. Values represent the mean ± SEM from three independent experiments. ##p < 0.01 compared with the no-treatment condition; *p < 0.05 and **p < 0.01 compared with the TNF-α/IFN-γ treatment condition.
All other figures should be labelled similarly.
All other corrections are completely satisfactory to me.
Reply: Thank you for your valuable feedback and for pointing out the need for proper figure captions. We have carefully reviewed your comments and modified the captions of all figures to align with the example provided for Figure 2. Each caption now provides a general overview of the information presented in the respective figures, as per your guidance.
Reviewer 3 Report
Comments and Suggestions for Authors
Please check the size and bold of the letters in all figures and revise them accordingly. Other than that, I think it will be well-revised and ready for publication.
Author Response
Please check the size and bold of the letters in all figures and revise them accordingly. Other than that, I think it will be well-revised and ready for publication.
Reply: Thank you for your feedback. We have carefully reviewed the size and boldness of the letters in all figures and revised them accordingly. We appreciate your thorough review and are pleased that the manuscript is now well-revised and ready for publication.